# Manifold Alignment across Geometric Spaces for Knowledge Base Representation Learning

**Huiru Xiao**                                                                  HXIAOAF@CSE.UST.HK
**Yangqiu Song**                                                                YQSONG@CSE.UST.HK
*Hong Kong University of Science and Technology*

## Abstract

Knowledge bases have multi-relations with distinctive properties. Most properties such as symmetry, inversion, and composition can be handled by the Euclidean embedding models. Nevertheless, transitivity is a special property that cannot be modeled efficiently in the Euclidean space. Instead, the hyperbolic space characterizes the transitivity naturally because of its tree-like properties. However, the hyperbolic space reveals its weakness for other relations. Therefore, building a representation learning framework for all relation properties is highly difficult. In this paper, we propose to learn the knowledge base embeddings in different geometric spaces and apply manifold alignment to align the shared entities. The aligned embeddings are evaluated on the out-of-taxonomy entity typing task, where we aim to predict the types of the entities from the knowledge graph. Experimental results on two datasets based on YAGO3 demonstrate that our approach has significantly good performances, especially in low dimensions and on small training rates.

## 1. Introduction

Representation learning plays an important role in the knowledge base (or in general multi-relational database) inference as well as its downstream tasks [Nickel et al., 2016]. Relations in knowledge bases have distinctive properties, such as symmetry, inversion, and composition. Transitivity is a typical property in knowledge bases. Transitive relations such as *IsA* are commonly used in many popular knowledge bases with taxonomies, such as YAGO [Suchanek et al., 2007] and WordNet [Miller, 1995].

Transitive relation has been shown quite different from other relations. While symmetry, inversion, and composition can be easily handled by the Euclidean space [Sun et al., 2019], the Euclidean embedding methods suffer from severe limitations for transitive relations and tree-like structures [Linial et al., 1995]. In contrast, the hyperbolic space is capable of embedding any finite tree while preserving the distances approximately [Gromov, 1987], so the transitivity is naturally characterized by the hyperbolic space [Nickel and Kiela, 2017].

However, the hyperbolic space cannot achieve promising results on capturing various relation patterns [Kolyvakis et al., 2019, Balazevic et al., 2019, Chami et al., 2020] due to the incompatibility between the geometry and general graph structures. To learn the embeddings of a wide variety of structures, [Gu et al., 2019] proposed to learn graph embeddings in a product manifold combining several spaces. Nevertheless, the model also focuses on single-relation graphs, making it inapplicable to knowledge bases.

Therefore, it is difficult to find a unified space to characterize all relation properties because both the Euclidean and the hyperbolic embeddings tackle some relation properties while having weaknesses on others. Hence, it is natural to think about an alternative

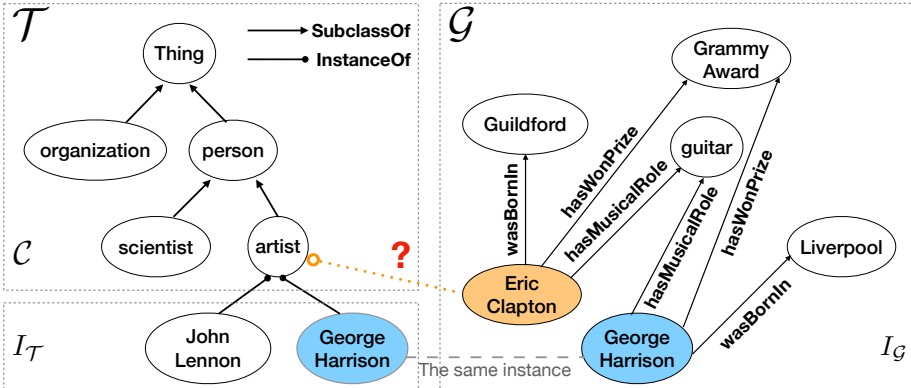

Figure 1: A simplified example of the knowledge base, which can be divided into the taxonomy $\mathcal{T}$ and the knowledge graph $\mathcal{G}$. The taxonomy has two transitive relations (*SubClassOf* and *InstanceOf*) while the knowledge graph has multi-relations without transitivity. *George Harrison* is an intersection entity between $\mathcal{T}$ and $\mathcal{G}$. Our task is to predict the type for the out-of-taxonomy entity *Eric Clapton*.

approach that uses different geometric spaces to best characterize various relation properties, while aligning the shared entities.

In this paper, we make use of manifold learning to address the issue. We propose to embed the knowledge base in different spaces according to the relation properties and then align the embeddings based on the local structures as well as the correspondences by manifold alignment [Ma and Fu, 2011], where the intersection entities act as the correspondences across spaces. *Manifold* is a topological space that is locally Euclidean, i.e., around every point, there is a neighborhood that is topologically the same as the open unit ball in Euclidean space. Although the hyperbolic geometry differs a lot from Euclidean geometry, the hyperbolic manifold is locally Euclidean. We use the projections between the hyperbolic and the Euclidean embeddings to align the underlying local structures of the knowledge base. Through manifold alignment, we can recover the correspondences across the sub-graphs as well as preserving their local structures.

We evaluate the aligned embeddings through an out-of-taxonomy entity typing task: given a knowledge base containing the taxonomy and the knowledge graph with their intersection entities, as Figure 1 shows, the goal is to predict the types of the out-of-taxonomy entities from the knowledge graph. The task is important in practice since some entities are missing in real-world taxonomies but occur in the knowledge graphs. Manual labeling is unpractical because of the large number of new entities and the high labeling cost. In the empirical evaluation, our approach achieves significant improvements. Notably, our approach has superior performances even when the training taxonomy is very small.

The code and data of our work are available at https://github.com/HKUST-KnowComp/GeoAlign.

## 2. Related Work

**Knowledge graph embeddings.** Traditional knowledge graph embedding models can be translation-based [Bordes et al., 2013] or bilinear models [Nickel et al., 2011, Yang et al., 2015]. Several extensions have been made in the real Euclidean space [Lin et al., 2015, Ji

et al., 2015, Sun et al., 2019] and the complex Euclidean space [Trouillon et al., 2016, Zhang et al., 2019]. It is shown that several properties of relations such as symmetry, inversion, and composition can be well handled by the Euclidean space [Sun et al., 2019].

**Hierarchy-aware knowledge graph embeddings.** Some works made special efforts for transitive relations in the Euclidean space. TransC [Lv et al., 2018] encoded types as spheres and entities as vectors for modeling the hypernymy relation HAKE [Zhang et al., 2020] proposed to map entities into the Euclidean polar coordinate system. JOIE [Hao et al., 2019] separated the knowledge base into the taxonomy and the remaining knowledge graph, which had the same setting as our work. It then leveraged a non-linear transformation between two Euclidean embedding models. However, the non-linear affine transformation is not powerful enough to build correlations between two underlying structures.

**Hyperbolic embeddings.** Hyperbolic embeddings have gained much attention in recent years. [Nickel and Kiela, 2017] proposed to use the Poncaré ball model of the hyperbolic space to learn the graph embeddings. [Ganea et al., 2018, Nickel and Kiela, 2018, Bécigneul and Ganea, 2019, Sala et al., 2018, Gu et al., 2019, Sonthalia and Gilbert, 2020] then further improved hyperbolic embeddings. These methods perform well on data with hierarchical structures and single transitive relation, but they cannot predict the out-of-taxonomy entities, thus are not directly applicable to our entity typing task. Motivated by the above works, MurP [Balazevic et al., 2019], AttH [Chami et al., 2020], and HyperKA [Sun et al., 2020] explored the multi-relational knowledge graph embeddings in the hyperbolic space. However, most relations in knowledge graphs do not have transitivity, thus do not fit the hyperbolic space. Their experimental results only significantly improved on transitive relations or the datasets with natural hierarchical structures.

**Manifold alignment.** Manifold alignment [Ham et al., 2003, Ma and Fu, 2011] is a class of algorithms aligning the local structures and transferring knowledge across data sets. In this work, we use two-step alignment [Lafon et al., 2006, Wang and Mahadevan, 2008], which utilizes the Laplacian eigenmaps [Belkin and Niyogi, 2003].

Note that our work focuses on a different research topic with the knowledge graph entity alignment [Hao et al., 2016, Sun et al., 2020] and the ontology matching [Alvarez-Melis et al., 2020] since entity alignment/matching aligns the entities that referring to the same thing but having different names, while in our work, the correspondences between the knowledge graph and the taxonomy are already known and we make use of the correspondences to apply manifold alignment and then predict the new out-of-taxonomy entities' types.

## 3. Problem Formulation

Given a knowledge base $\mathcal{B}$ containing the taxonomy $\mathcal{T}$ and the knowledge graph $\mathcal{G}$, we aim to learn the alignment between the embeddings of $\mathcal{T}$ and $\mathcal{G}$. An example of $\mathcal{B}$ is shown in Figure 1. The taxonomy $\mathcal{T}$ contains the type set $\mathcal{C} = \{Thing, organization, person, scientist, artist\}$, the entity set $I_{\mathcal{T}} = \{John\ Lennon, George\ Harrison\}$, and the directed edge set $E_{\mathcal{T}} = \{e_{i,j} : i, j \in \mathcal{C} \cup I_{\mathcal{T}}\}$, where $e_{i,j}$ represents that $i$ and $j$ have the transitive relation, e.g., (*John Lennon, InstanceOf, artist*) and (*artist, SubClassOf, person*).

Denote the entity set and the relation set of the knowledge graph $\mathcal{G}$ as $I_\mathcal{G}$ and $R_\mathcal{G}$ respectively,[1] then $\mathcal{G}$ is composed of the triplets $F = \{(h, r, t) : h, t \in I_\mathcal{G}, r \in R_\mathcal{G}\}$, e.g., (*Eric Clapton, wasBornIn, Guildford*). There are some intersection entities between $\mathcal{T}$ and $\mathcal{G}$, which we denote as $I_{cor}$: $I_{cor} = I_\mathcal{T} \cap I_\mathcal{G}$. In Figure 1, $I_{cor} = \{George\ Harrison\}$.

To apply manifold alignment, first, we need to obtain the pretrained embeddings of $I_\mathcal{T}$ and $I_\mathcal{G}$, denoted as $\mathbf{I}^\mathcal{T}$ and $\mathbf{I}^\mathcal{G}$. $\mathbf{I}^\mathcal{T} \in \mathbb{H}^n$, $\mathbf{I}^\mathcal{G} \in \mathbb{R}^m$, where $\mathbb{H}^n$ is the $n$-dimensional hyperbolic space, and $\mathbb{R}^m$ is the $m$-dimensional Euclidean space. Then manifold alignment is applied to project $\mathbf{I}^\mathcal{T}$ and $\mathbf{I}^\mathcal{G}$ into a shared manifold. After that, we have the embeddings of all entities of $\mathcal{B}$ in the same manifold. Note that here all entities do not include types in $\mathcal{C}$ in the taxonomy. In the out-of-taxonomy entity typing task, the goal is to predict the types of the new entities from $I_n = I_\mathcal{G} - I_\mathcal{T}$. It can also be regarded as a taxonomy completion task.

## 4. Approach

The main framework of our approach involves three parts: pretraining, manifold alignment, and retraining the taxonomy with new entities. We give the details in the following.

### 4.1 Pretraining of Taxonomy and Knowledge Graph Embeddings

In our work, we use the hyperboloid model of hyperbolic embeddings [Nickel and Kiela, 2018] to learn the taxonomy embeddings. The background of the hyperbolic space and the hyperboloid model are in Appendix A. For the knowledge graph, we employ TransE proposed by [Bordes et al., 2013], which is a classical and effective embedding algorithm.

#### 4.1.1 HYPERBOLIC EMBEDDINGS IN THE HYPERBOLOID MODEL

Given the taxonomy $\mathcal{T}$ with type set $\mathcal{C}$, entity set $I_\mathcal{T}$, and edge set $E_\mathcal{T}$, the objective is to find the embeddings of the types and entities $\mathbf{T} = \{\mathbf{T}_i\}, \mathbf{T}_i \in \mathcal{H}^n$, where $\mathcal{H}^n$ is the $n$-dimensional hyperboloid model ($\mathcal{H}^n$ is one model of $\mathbb{H}^n$). The soft ranking loss is

$$\mathcal{L}_\mathcal{T} = \sum_{(x_i, x_j) \in E_\mathcal{T}} \log \frac{e^{-d_h(\mathbf{T}_i, \mathbf{T}_j)}}{\sum_{x_k \in \mathcal{N}(x_i)} e^{-d_h(\mathbf{T}_i, \mathbf{T}_k)}}, \tag{1}$$

where $\mathcal{N}(x_i) = \{x_k : (x_i, x_k) \notin E_\mathcal{T}\} \cup \{x_i\}$ is the set of negative examples for $x_i$ together with $x_i$. The hyperboloid model $\mathcal{H}^n$ and its distance function $d_h$ are defined in Appendix A.

The minimization of $\mathcal{L}_\mathcal{T}$ makes the connected entities and types closer than those with no observed edges. Riemannian SGD (RSGD) [Bonnabel, 2013] is applied to train the hyperbolic embeddings.

#### 4.1.2 TRANSE FOR KNOWLEDGE GRAPH EMBEDDINGS

Given a triplet $(h, r, t)$, by regarding $d(\mathbf{h} + \mathbf{r}, \mathbf{t})$ as the energy of $(h, r, t)$, where $\mathbf{h}, \mathbf{r}, \mathbf{t}$ are the corresponding embeddings and $d$ is some dissimilarity measure (usually the $L_1$ or $L_2$

---

1. We suppose the knowledge graph $\mathcal{G}$ does not have transitive relations since all edges with transitive relations are in the taxonomy.

norm), the margin-based ranking loss over the knowledge graph $\mathcal{G}$ with the triplet set $F$ is

$$\mathcal{L}_{\mathcal{G}} = \sum_{(h,r,t)\in F} \sum_{(h',r,t')\in\mathcal{N}(h,r,t)} [\gamma + d(\mathbf{h}+\mathbf{r},\mathbf{t}) - d(\mathbf{h}'+\mathbf{r},\mathbf{t}')]_+, \quad (2)$$

for some margin $\gamma > 0$. $[x]_+ = \max(0, x)$ and $\mathcal{N}(h, r, t) = \{(h', r, t)|h' \in I_{\mathcal{G}}\} \cup \{(h, r, t')|t' \in I_{\mathcal{G}}\}$ is the negative sample set. The optimization of $\mathcal{L}_{\mathcal{G}}$ encourages positive triplets to satisfy $\mathbf{h} + \mathbf{r} \approx \mathbf{t}$ and negative ones to satisfy that $\mathbf{h} + \mathbf{r}$ is far away from $\mathbf{t}$.

### 4.2 Manifold Alignment

After pretraining, we conduct manifold alignment to learn the aligned embeddings of the taxonomy entities $I_{\mathcal{T}}$ and knowledge graph entities $I_{\mathcal{G}}$ in a shared manifold. The goal of manifold alignment is to find the projection functions such that the projections not only minimize the distance between the corresponding points but also preserve the local manifold structures of the original data. In our framework, given two entity sets $I_{\mathcal{T}}$, $I_{\mathcal{G}}$, and their embeddings $\mathbf{I}^{\mathcal{T}} \in \mathbb{H}^n$, $\mathbf{I}^{\mathcal{G}} \in \mathbb{R}^m$, we aim to learn the projections $\phi_{I_{\mathcal{T}}} : \mathbb{H}^n \to \mathbb{R}^d$ and $\phi_{I_{\mathcal{G}}} : \mathbb{R}^m \to \mathbb{R}^d$. The algorithm works as follows:

First, we construct the binary correspondence matrix $W$ using the intersection set $I_{cor}$:

$$W_{ij} = \begin{cases} 1 & if\ (I_{\mathcal{T}})_i \leftrightarrow (I_{\mathcal{G}})_j, \\ 0 & otherwise. \end{cases} \quad (3)$$

Next, we construct the adjacency graphs from $\mathbf{I}^{\mathcal{T}}$ and $\mathbf{I}^{\mathcal{G}}$. Specifically, we compute the pointwise distances within the two sets and select the $k$ nearest neighbors ($k$-nn) to construct the adjacency matrices $A^{\mathcal{T}}$ and $A^{\mathcal{G}}$, i.e., $A_{ij}^{\mathcal{T}} = 1$ if and only if entity $i$ (or $j$) is among the $k$ nearest neighbors of entity $j$ (or $i$), and the same with $A^{\mathcal{G}}$. Then we construct the similarity matrices $S^{\mathcal{T}}$ and $S^{\mathcal{G}}$ using heat kernel:

$$S_{ij}^{\mathcal{T}} = \exp(-\frac{d_h(\mathbf{I}_i^{\mathcal{T}}, \mathbf{I}_j^{\mathcal{T}})}{t}) \cdot A_{ij}^{\mathcal{T}}, \qquad S_{ij}^{\mathcal{G}} = \exp(-\frac{d(\mathbf{I}_i^{\mathcal{G}}, \mathbf{I}_j^{\mathcal{G}})}{t}) \cdot A_{ij}^{\mathcal{G}}, \quad (4)$$

where $t > 0$ is the parameter of heat kernel. $d_h$ refers to the hyperboloid distance in Appendix A, and $d$ is the $L_2$ norm in the Euclidean space. The manifold alignment loss is defined as:

$$\mathcal{L}_M = \mu \left[ \sum_{i,j} \|\phi_{I_{\mathcal{T}}}(\mathbf{I}_i^{\mathcal{T}}) - \phi_{I_{\mathcal{T}}}(\mathbf{I}_j^{\mathcal{T}})\|^2 S_{ij}^{\mathcal{T}} + \sum_{i,j} \|\phi_{I_{\mathcal{G}}}(\mathbf{I}_i^{\mathcal{G}}) - \phi_{I_{\mathcal{G}}}(\mathbf{I}_j^{\mathcal{G}})\|^2 S_{ij}^{\mathcal{G}} \right]$$
$$+ (1-\mu) \sum_{i,j} \|\phi_{I_{\mathcal{T}}}(\mathbf{I}_i^{\mathcal{T}}) - \phi_{I_{\mathcal{G}}}(\mathbf{I}_j^{\mathcal{G}})\|^2 W_{ij}, \quad (5)$$

where $0 \le \mu \le 1$ is the weight to balance between preserving the original manifold structures and minimizing the corresponding entity distances. The first term multiplied by $\mu$ infer that if two entities both from $I_{\mathcal{T}}$ or both from $I_{\mathcal{G}}$ are similar, their projections in the latent space should be close with each other. If two entities from $I_{\mathcal{T}}$ and $I_{\mathcal{G}}$ are the same entity, the last term minimizes the distance between their projections in the shared manifold.

In practice, an additional constraint needs to be added in Eq. (5) to avoid the all-zero solution. More details are given in Appendix B.1. Then minimizing $\mathcal{L}_M$ is equivalent to solving a generalized eigenvalue problem of the joint graph Laplacian $L$: $L\mathbf{v} = \lambda D\mathbf{v}$.

Denote the joint matrix $J = \begin{bmatrix} \mu S^{\mathcal{T}} & (1-\mu)W \\ (1-\mu)W^T & \mu S^{\mathcal{G}} \end{bmatrix}$, $D$ is a diagonal matrix and $D_{ii} = \sum_j J_{ji} = \sum_j J_{ij}$ (the proof of $J$'s symmetry can be referred to Appendix B.2), then the Laplacian $L = D - J$ is a symmetric and positive semidefinite matrix (see Appendix B.3 for proof). Let $\mathbf{v}_0, \ldots, \mathbf{v}_d$ be the solutions ordered according to their eigenvalues, i.e., $L\mathbf{v}_i = \lambda_i D\mathbf{v}_i$ for $0 \leq i \leq d$, and $0 = \lambda_0 \leq \lambda_1 \leq \cdots \leq \lambda_d$. Then the trivial eigenvector $\mathbf{v}_0$ is discarded and the next $d$ eigenvectors are used for the aligned embeddings (proved in Appendix B.4). That means the embedding vectors for $I_{\mathcal{T}}$ and $I_{\mathcal{G}}$ in the shared manifold are

$$\phi_{I_{\mathcal{T}}}(\mathbf{I}_i^{\mathcal{T}}) = (\mathbf{v}_1(i), \ldots, \mathbf{v}_d(i)), \quad 1 \leq i \leq |I_{\mathcal{T}}|, \tag{6}$$

$$\phi_{I_{\mathcal{G}}}(\mathbf{I}_j^{\mathcal{G}}) = (\mathbf{v}_1(|I_{\mathcal{T}}| + j), \ldots, \mathbf{v}_d(|I_{\mathcal{T}}| + j)), \ 1 \leq j \leq |I_{\mathcal{G}}|. \tag{7}$$

Following the above procedure, we project both entity sets into a shared manifold. The aligned embeddings are learned by preserving the original manifold structure and recovering the correspondences. They provide correlations between $I_{\mathcal{T}}$ and $I_{\mathcal{G}}$.

## 4.3 Linking New Entities and Retraining

Once we obtain the aligned embeddings $\phi_{I_{\mathcal{T}}}(\mathbf{I}^{\mathcal{T}}), \phi_{I_{\mathcal{G}}}(\mathbf{I}^{\mathcal{G}}) \in \mathbb{R}^d$, we can compute the pairwise distances of the entities in the shared manifold. From the pairwise distances, we further connect $I_{\mathcal{T}}$ and $I_n = I_{\mathcal{G}} - I_{\mathcal{T}}$ by $k$-nn and thus have the new edges $E_n = \{(x_i, x_j) : x_i \in I_{\mathcal{T}}, x_j \in I_n\}$.

Next, we retrain the hyperbolic embeddings of the completed taxonomy with $E_{\mathcal{T}} \cup E_n$. Note that $E_n$ is different with $E_{\mathcal{T}}$ since $E_{\mathcal{T}}$ is the set of original taxonomy edges representing the transitive relation among types and entities while edges in $E_n$ reveal the similarities between entities. To differentiate them, we regard the edges in $E_n$ as weighted undirected edges during training. That is to say, we append $(x_i, x_j)$ and $(x_j, x_i)$ to the training set if $(x_i, x_j) \in E_n$. Moreover, we let the terms associating with $E_n$ in the loss (Eq. (1)) multiply by a weight, which adjusts the weight of the newly-added undirected edges.

Again we use RSGD to update the hyperbolic embeddings. Then we can predict the links between the new entities $I_n$ and the types $\mathcal{C}$ according to the hyperboloid distance.

## 5. Experiments

In this section, we evaluate the performance of our approach on the out-of-taxonomy entity typing task. We report the main results here. For more experiments, please see Appendix D.

## 5.1 Experimental Settings

### 5.1.1 DATA

We construct our knowledge bases from YAGO3 [Mahdisoltani et al., 2015], a huge semantic knowledge base derived from Wikipedia, WordNet, and GeoNames. Consistent with our framework, YAGO3 is divided into the taxonomy and the knowledge graph. We provide the

|  | Taxonomy | | KG |
|---|---|---|---|
|  | YAGOwordnet | wikiObjects | YAGOfacts |
| Depth/# Relations | 13 | 16 | 37 |
| # Types | 233 | 3,369 | 0 |
| # Entities | 8,927 | 14,006 | 50,566 |
| # Intersection | 8,608 | 13,880 | - |
| # Edges/Triplets | 92,479 | 153,643 | 392,335 |
| Training/Test Entities | 4,464/4,463 | 7,003/7,003 | 50,566/- |
| Training/Test Edges | 47,038/45,441 | 91,496/62,147 | 392,335/- |
| $\delta$-hyperbolicity | 0.5 | 0.5 | 1.5 |

Table 1: Taxonomy/Knowledge graph statistics. Depth is for taxonomy while # Relations is for KG. # Intersection refers to the number of intersection entities between the taxonomy and YAGOfacts. We ignore the edge attributes (i.e. multi-relations) when computing $\delta$-hyperbolicity for YAGOfacts.

statistics of our datasets in Table 1. We also give their Gromov's $\delta$-hyperbolicity [Gromov, 1987], which measures the tree-likeness of graphs (refer to Appendix C for definition). The lower $\delta$ corresponds to the more tree-like graph. We sample our datasets as follows:

**YAGOwordnet.** In YAGO3, the taxonomy is formed with *yago-types* and *yago-taxonomy*. They include relations *SubClassOf* and *type*. The types are from Wikipedia and WordNet. We extract all WordNet types and sample around 9K entities from *yago-types* and *yago-taxonomy*, then trace back from the sampled entities until the root to construct a hierarchy. As in previous works [Nickel and Kiela, 2017, 2018], we compute the transitive closure of the hierarchy to construct the YAGOwordnet taxonomy.

**wikiObjects.** From *yago-types* and *yago-taxonomy*, we extract all Wikipedia types and their descendants (including types and entities) to obtain a forest, where each tree has a root in Wikipedia types. We then sample a subtree from the root $\langle wikicat\_Objects \rangle$ and compute its transitive closure to construct the wikiObjects taxonomy.

**YAGOfacts.** In YAGO3, *yago-facts* is the core knowledge graph which contains all facts (triplets) of YAGO3 that hold among entities. Different with *yago-types* and *yago-taxonomy*, *yago-facts* does not contain the transitive relations or any type. We select around 50K most frequent entities from *yago-facts* and extract all triplets whose head entity and tail entity both belong to the selected entities to form YAGOfacts.

For each taxonomy (YAGOwordnet/wikiObjects), we randomly split the entities into training entities (50%) and test entities (50%) with the constraint that the test entities must occur in YAGOfacts, otherwise, YAGOfacts cannot provide any reference about the test entities. Then we discard all edges containing any test entity to obtain the training taxonomy. The training taxonomy and YAGOfacts are used for pretraining and the embeddings of the training entities are used for manifold alignment with embeddings of YAGOfacts' entities. The evaluation task is to predict the types of the out-of-taxonomy test entities. The training taxonomy corresponds to $\mathcal{T}$ in Section 3 while YAGOfacts corresponds to $\mathcal{G}$.

### 5.1.2 Baselines

The following baselines are compared with our approach (**GeoAlign**): KG embedding models, including **TransE** [Bordes et al., 2013], **ComplEx** [Trouillon et al., 2016], **RotatE** [Sun

et al., 2019]; hierarchy-aware Euclidean methods, including **TransC** [Lv et al., 2018], **HAKE** [Zhang et al., 2020], **JOIE** [Hao et al., 2019]; multi-relational hyperbolic models: **MurP** [Balazevic et al., 2019], **AttH** [Chami et al., 2020], **HyperKA** [Sun et al., 2020].

### 5.1.3 TRAINING AND EVALUATION

To apply the baselines to our task, they are trained on all triplets of YAGOfacts combined with the training taxonomy, where the taxonomy edges are labeled as $\langle isA \rangle$. Then the test triplets are $\{(x_i, \langle isA \rangle, \mathcal{C}_j)\}$ where $x_i$ is a test entity and $\mathcal{C}_j$ is $x_i$'s ground-truth type.

For knowledge graph embedding models, we use the OpenKE repository [Han et al., 2018] to train them while for other baselines, we use their public codes. For all methods, we tune the hyperparameters on the knowledge base combining YAGOwordnet and YAGOfacts by grid search according to MAP score. The hyperparameters are given in Appendix D.1.

We use the mean average precision (MAP), mean reciprocal rank (MRR), and the proportion of correct types that rank no larger than N (Hits@N) as our evaluation metrics, which are widely used for evaluating ranking and link prediction. The details of prediction steps and the evaluation metrics are given in Appendix D.2. In our experiments, each running is executed 5 times and the mean values of results are reported.

## 5.2 Overall Results

Table 2 presents the results in 50-dimensional embedding spaces. The results show that GeoAlign has the best performance on wikiObjects while having a very close performance with MurP on YAGOwordnet. In fact, wikiObjects is more challenging since its taxonomy is more massive (see Appendix D.3 for the case study), making it more difficult to find all correct types for the out-of-taxonomy entity.

From Table 2, we see that the traditional Euclidean models (TransE, ComplEx, and RotatE) are not capable of inferring the transitive relation, which is consistent with previous works on hyperbolic embeddings [Nickel and Kiela, 2017, 2018]. The hierarchy-aware methods (TransC, HAKE, and JOIE) have better results than the traditional Euclidean embeddings, but overall they cannot achieve comparative performances with the hyperbolic models.

For the multi-relational hyperbolic models, MurP and AttH, which use different relation parameterizations on the base of Poincaré embeddings, reveal their strengths. MurP, AttH, and GeoAlign all take advantage of the hyperbolic geometry for the taxonomy embeddings, thus having close results on the entity typing task. However, in Section 5.4, we will show that GeoAlign has significant improvements over MurP and AttH when the training taxonomy is very small. Furthermore, MurP and AttH can only do the inference for all relation properties in the hyperbolic space, which is not suitable for the non-transitive relation properties, while GeoAlign can take advantage of any base embedding models rather than TransE for pretraining the knowledge graph. As for HyperKA, which leverages hyperbolic GNN for embeddings, we tried our best to tune the model, but it still cannot achieve promising results. We think the neural models may not fit this task setting and our datasets. That also accounts for why AttH is not as good as MurP. Compared with MurP, AttH adds the attention mechanism and has more complicated parameterizations. It may hurt the model's feasibility sometimes.

| | YAGOwordnet | | | | wikiObjects | | | |
|---|---|---|---|---|---|---|---|---|
| | MAP | MRR | Hits@1 | Hits@3 | MAP | MRR | Hits@1 | Hits@3 |
| TransE | ‡21.36 | ‡6.91 | ‡19.67 | ‡27.95 | ‡13.68 | ‡6.51 | ‡14.52 | ‡24.10 |
| ComplEx | ‡50.77 | ‡13.63 | ‡20.09 | ‡48.82 | ‡28.10 | ‡15.13 | ‡24.30 | ‡33.89 |
| RotatE | ‡70.72 | ‡21.63 | ‡64.63 | ‡88.28 | ‡62.42 | ‡32.38 | ‡66.85 | ‡81.78 |
| TransC | ‡84.25 | ‡26.35 | ‡93.72 | †99.29 | ‡67.87 | ‡34.24 | ‡81.87 | ‡93.26 |
| HAKE | ‡74.52 | ‡18.77 | ‡51.22 | ‡58.86 | †66.50 | †30.53 | ‡52.93 | ‡61.02 |
| JOIE | **94.92** | 28.15 | ‡97.20 | 99.60 | ‡86.67 | ‡41.87 | ‡93.00 | ‡98.94 |
| MurP | †94.41 | **28.20** | **99.57** | †99.84 | ‡_88.48_ | ‡_42.77_ | ‡_99.80_ | **100.00** |
| AttH | _94.80_ | _28.19_ | ‡97.76 | _99.88_ | 88.40 | ‡42.65 | ‡98.34 | †99.90 |
| HyperKA | ‡55.83 | ‡19.15 | ‡64.84 | ‡83.84 | ‡48.02 | ‡27.42 | ‡59.80 | ‡74.73 |
| GeoAlign | ‡94.04 | _28.19_ | †_99.13_ | **99.92** | **88.67** | **42.82** | **99.89** | **100.00** |

Table 2: Results of MAP(%), MRR(%), and Hits@N(%) in 50-dimensional embedding spaces. The best results are shown in boldface and the second-best results are underlined. The statistical significance metrics are marked with either † if p-values < 0.05 or ‡ if p-values < 0.001.

| Dimension | 5 | | 10 | | 20 | | 100 | |
|---|---|---|---|---|---|---|---|---|
| | MRR | Hits@1 | MRR | Hits@1 | MRR | Hits@1 | MRR | Hits@1 |
| TransE | ‡2.42 | ‡3.91 | ‡5.41 | ‡9.91 | ‡6.80 | ‡14.14 | ‡8.77 | ‡19.21 |
| ComplEx | ‡0.09 | ‡0.00 | ‡11.97 | ‡22.59 | ‡14.32 | ‡24.20 | ‡20.21 | ‡34.39 |
| RotatE | ‡1.76 | ‡1.39 | ‡3.50 | ‡2.94 | ‡4.23 | ‡3.37 | ‡37.68 | ‡79.95 |
| TransC | ‡32.62 | ‡66.69 | ‡36.15 | ‡78.55 | ‡36.67 | ‡84.03 | ‡34.28 | ‡81.32 |
| HAKE | ‡7.85 | ‡12.62 | ‡15.58 | ‡22.70 | ‡10.16 | ‡17.29 | †37.90 | †70.88 |
| JOIE | ‡27.76 | ‡48.01 | ‡39.93 | ‡84.68 | ‡41.64 | ‡92.62 | ‡42.12 | ‡94.61 |
| MurP | ‡_42.18_ | ‡_98.19_ | ‡_42.61_ | ‡_98.59_ | ‡_42.72_ | ‡_99.60_ | **42.91** | ‡_99.90_ |
| AttH | - | - | †40.60 | †92.47 | ‡42.10 | ‡97.05 | ‡42.78 | ‡98.76 |
| HyperKA | ‡16.93 | ‡34.51 | ‡21.65 | ‡41.30 | ‡19.00 | ‡42.98 | ‡24.51 | ‡61.92 |
| GeoAlign | **42.74** | **99.64** | **42.83** | **99.84** | **42.81** | **99.92** | ‡_42.82_ | **99.93** |

Table 3: Results of MRR(%) and Hits@1(%) in different embedding dimensions on wikiObjects. AttH requires the dimension to be even because of the diagonal Givens transformations in its model, thus not applicable to 5-dimensional space. The best results are shown in boldface and the second-best results are underlined. The statistical significance metrics are marked with either † if p-values < 0.05 or ‡ if p-values < 0.001.

## 5.3 Exploring the Embedding Dimensions

In this section, we explore the performances in different embedding dimensions. The results are presented in Table 3. For GeoAlign, JOIE, and HyperKA, the embedding dimensions for the knowledge graph and the taxonomy are both set as $n$ for $n \in \{5, 10, 20, 100\}$. From Table 3, we see that with the increase of the embedding dimension, most methods get better results. The Euclidean models can have big improvements in higher dimensions, such as RotatE from 20-d to 100-d, but their 100-d performances cannot surpass GeoAlign in 5-d. We also notice that some methods suffer from overfitting in high dimensions, e.g., TransC, HAKE, and HyperKA drop down when the dimension increases. In contrast, GeoAlign, MurP, and AttH achieve great results steadily. On the one hand, the hyperbolic models require much lower dimensions. On the other hand, 5-dimensional embeddings are already

| Training rate | 0.1 | 0.2 | 0.3 | 0.4 |
|---|---|---|---|---|
| Training/Test entities | 1,401/12,605 | 2,802/11,204 | 4,202/9,804 | 5,603/8,403 |
| Training/Test edges | 42,764/110,879 | 54,637/99,006 | 67,788/85,855 | 80,190/73,453 |
| MurP | ‡82.83 | ‡86.56 | ‡88.16 | ‡88.45 |
| AttH | †84.32 | ‡86.62 | ‡87.80 | 88.40 |
| GeoAlign | **88.50** | **88.78** | **89.09** | **88.90** |

Table 4: Results of MAP(%) under different training rates on wikiObjects in 50-dimension. The training rate is used to randomly split the taxonomy entities. Training edges represents the number of training edges in the training taxonomy wikiObjects (the training edges of YAGOfacts is 392,335 all the time). The best results are shown in boldface. The statistical significance metrics are marked with either † if p-values $< 0.05$ or ‡ if p-values $< 0.001$.

enough for GeoAlign to learn the manifold alignment between the local structures of the knowledge base.

## 5.4 Results on Small Training Rates

In Section 5.2, we see that the performances of MurP, AttH, and GeoAlign are very close under the training rate=0.5. Here we explore their performances on smaller training rates. For the training rate $r \in \{0.1, 0.2, 0.3, 0.4\}$, we randomly split the entities into training entities ($r$) and test entities ($1 - r$). The splitting and training taxonomy construction procedure are the same as described in Section 5.1.1. Note that the small training rate means that the pretrained taxonomy and the number of taxonomy entities used for manifold alignment are both small. We report the MAP(%) scores in Table 4. We find that when the training rate is small, GeoAlign outperforms MurP and AttH significantly. With the increase of the training rates, their performances get more and more similar and converge to stable. The results demonstrate the effectiveness of our approach on small training rates.

## 5.5 Ablation Study

### 5.5.1 ON THE RETRAINING STEP

To analyze the benefits and the potential defects of the retraining step after manifold alignment, we compare the performances of GeoAlign with and without retraining on two tasks. The first task is the out-of-taxonomy entity typing task, which is the same as the above experiments. The -w/o retraining method works in the way that after manifold alignment, we directly predict the types of the test entities as the types of its nearest neighbor according to the aligned embeddings. If its nearest neighbor is not in the taxonomy's training entities, we find the next nearest one until it does. The second task is graph reconstruction of the training taxonomy. We intend to see whether the retrained model with new entities compromises the representation of the original taxonomy. We report the results in Table 5.

The entity typing results demonstrate that after manifold alignment, the retraining of the embeddings with the added edges incorporates more structural information of the taxonomy with new entities and improves the performances significantly. The reconstruction results show that the retraining has little impact on the original training taxonomy. The compromise is acceptable, especially when considering the remarkable improvements on the entity typing task (more than 10% MAP improvements in Table 5).

|  |  | YAGOwordnet | | wikiObjects | |
|---|---|---|---|---|---|
|  |  | MAP | MRR | MAP | MRR |
| Entity typing | GeoAlign | 94.04 | 28.19 | 88.67 | 42.82 |
|  | -w/o retraining | 84.98 | - | 71.75 | - |
| Reconstruction | GeoAlign | 95.40 | 28.70 | 89.69 | 39.17 |
|  | -w/o retraining | 97.06 | 29.02 | 90.99 | 39.56 |

Table 5: Ablation study for GeoAlign in 50-dimension. Entity typing is the out-of-taxonomy entity typing task and Reconstruction is the training taxonomy reconstruction task. Since -w/o retraining does not obtain a rank for all types on entity typing task, MRR is not applicable.

|  |  | MAP | MRR |
|---|---|---|---|
| Entity typing | GeoAlign-hyperboloid | 94.04 | 28.19 |
|  | GeoAlign-Poincaré | 84.30 | 26.56 |
| Reconstruction | GeoAlign-hyperboloid | 95.40 | 28.70 |
|  | GeoAlign-Poincaré | 91.68 | 28.11 |

Table 6: Ablation study of the hyperbolic models on YAGOwordnet in 50-dimension. Entity typing is the out-of-taxonomy entity typing and Reconstruction is the training taxonomy reconstruction.

### 5.5.2 ON THE HYPERBOLIC MODELS

To compare the different hyperbolic models, we evaluate the performances of GeoAlign with the hyperboloid model and the Poincaré ball model on two tasks. Again, the first task is the out-of-taxonomy entity typing task and the second task is graph reconstruction of the training taxonomy. The MAP(%) and MRR(%) scores are reported in Table 6. GeoAlign-hyperboloid (Poincaré) means we pretrain and retrain the taxonomy by the hyperboloid embeddings (Poincaré ball embeddings). From Table 6, we see that GeoAlign-hyperboloid surpasses GeoAlign-Poincaré a lot, especially on the out-of-taxonomy entity typing task.

## 6. Conclusion and Future Work

We propose to learn the embeddings of knowledge bases in different spaces and apply manifold alignment across the geometric spaces to build the projection. The main motivation is to allow different geometric spaces to model the various properties of relations as well as the various local structures of the knowledge base, while manifold alignment provides a way to incorporate the local manifold structure of two entity sets. We propose a solid framework and evaluate our approach on an out-of-taxonomy entity typing task. The empirical results demonstrate the superiority of our approach, especially in low dimensions and on small training rates. Future works include the exploration of broader types of geometries for learning embeddings and more effective approaches for aligning multiple manifolds.

## Acknowledgment

The authors of this paper were supported by the NSFC Fund (U20B2053) from the NSFC of China, the RIF (R6020-19 and R6021-20) and the GRF (16211520) from RGC of Hong Kong, the MHKJFS (MHP/001/19) from ITC of Hong Kong.

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

## A. Hyperbolic Space Background

Hyperbolic space is a homogeneous space with constant negative curvature. To describe hyperbolic space in mathematical language, there are five common-used models of hyperbolic space [Cannon et al., 1997], whose notations and definitions are as follows.

- The Half-space model $\mathcal{A}^n = \{(1, x_2, \ldots, x_{n+1}) : x_{n+1} > 0\}$.

- The Poincaré ball model $\mathcal{P}^n = \{(x_1, \ldots, x_n, 0) : x_1^2 + \cdots + x_n^2 < 1\}$.

- The Jemisphere model (or the Hemisphere model) $\mathcal{J}^n = \{(x_1, \ldots, x_{n+1}) : x_1^2 + \cdots + x_{n+1}^2 = 1, x_{n+1} > 0\}$.

- The Klein model $\mathcal{K}^n = \{(x_1, \ldots, x_n, 1) : x_1^2 + \cdots + x_n^2 < 1\}$.

- The hyperboloid model (or the Lorentz model) $\mathcal{H}^n = \{(x_1, \ldots, x_n, x_{n+1}) : x_1^2 + \cdots + x_n^2 - x_{n+1}^2 = -1, x_{n+1} > 0\}$.

The five models of n-dimensional hyperbolic space $\mathbb{H}^n$ are located in the ambient Euclidean space $\mathbb{R}^{n+1}$ and are isometric, i.e., for any two models $(\mathcal{X}, d)$ and $(\mathcal{X}', d')$, there exists a one-to-one mapping $f$ from $\mathcal{X}$ on to $\mathcal{X}'$ preserving all distances: $\forall x, y \in \mathcal{X}, d(x, y) = d'(f(x), f(y))$, and $f$ is called isometry.

For representation learning in the hyperbolic space, the Poincaré ball model [Nickel and Kiela, 2017] and the hyperboloid model [Nickel and Kiela, 2018] are the two most popular models due to their intuitive formulas and nice properties.

**Poincaré ball model [Nickel and Kiela, 2017].** Denote the Riemannian manifold of Poincaré ball model as $(\mathcal{P}^n, \rho_p)$, where $\mathcal{P}^n = \{\mathbf{x} \in \mathbb{R}^n : \|\mathbf{x}\| < 1\}$, which represents the open $n$-dimensional unit ball in Euclidean space ($\|\cdot\|$ is the Euclidean norm). $\rho_p$ is the metric tensor, given in Table 7.

**Hyperboloid model [Nickel and Kiela, 2018].** For the hyperboloid model $(\mathcal{H}^n, \rho_h)$, write $\mathcal{H}^n$ using Lorentzian inner product: $\mathcal{H}^n = \{\mathbf{x} \in \mathbb{R}^{n+1} : \langle \mathbf{x}, \mathbf{x} \rangle_{\mathcal{L}} = -1, x_0 > 0\}$, where $\langle \mathbf{x}, \mathbf{y} \rangle_{\mathcal{L}} = -x_0 y_0 + \sum_{i=1}^n x_i y_i$ is the Lorentzian inner product. The manifold is composed of points on the forward sheet of a two-sheeted hyperboloid. As mentioned before, Poincaré ball model and the hyperboloid model are isometric. The one-to-one mapping $f : \mathcal{H}^n \to \mathcal{P}^n$ is the central projection from the point $(0, \ldots, 0, -1)$:

$$f(x_0, x_1, \ldots, x_n) = \frac{x_1, \ldots, x_n}{x_0 + 1}, \tag{8}$$

$$f^{-1}(x_1, \ldots, x_n) = \frac{(1 + \|\mathbf{x}\|^2, 2x_1, \ldots, 2x_n)}{1 - \|\mathbf{x}\|^2}. \tag{9}$$

Both Poincaré ball model and the hyperboloid model are conformal models, i.e., they preserve angles of Euclidean space. Their metric tensors and distance functions are given in Table 7. The Poincaré distance has the property that when closer to the origin, it approximates to the Euclidean distance. Additionally, closer a point to the origin, the relatively smaller distances to other points it has. Correspondingly, the points near the boundary have very large distances with each other. Consider the tree structure, where the root node has relatively small distance with other nodes while leaf nodes are usually far away from each other. Moreover, the volume of a ball in the Poincaré model grows

| Model | Metric tensor | Distance function |
|---|---|---|
| Poincaré | $\rho_p(\mathbf{x}) = (\frac{2}{1-\|\mathbf{x}\|^2})^2 \rho_E(\mathbf{x})$ | $d_p(\mathbf{x}, \mathbf{y}) = arcosh\left(1 + 2\frac{\|\mathbf{x}-\mathbf{y}\|^2}{(1-\|\mathbf{x}\|^2)(1-\|\mathbf{y}\|^2)}\right)$ |
| Hyperboloid | $\rho_h(\mathbf{x}) = \begin{bmatrix} -1 & & & \\ & 1 & & \\ & & \ddots & \\ & & & 1 \end{bmatrix}$ | $d_h(\mathbf{x}, \mathbf{y}) = arcosh(-\langle \mathbf{x}, \mathbf{y} \rangle_{\mathcal{L}})$ |

Table 7: Metric tensors and distance functions of Poincaré ball model and the hyperboloid model. $\rho_E(\mathbf{x})$ is the Euclidean metric. $\langle \mathbf{x}, \mathbf{y} \rangle_{\mathcal{L}}$ is the Lorentzian inner product, $\langle \mathbf{x}, \mathbf{y} \rangle_{\mathcal{L}} = -x_0 y_0 + \sum_{i=1}^{n} x_i y_i$.

exponentially with its radius, resembling the tree-like property that the number of nodes grows exponentially with depth in a tree. [Krioukov et al., 2010] built the approximate equivalence between hierarchical networks and the hyperbolic space. Nevertheless, when using the Poincaré ball model for hierarchical structures [Nickel and Kiela, 2017], numerical instabilities arise, which motivates the use of the hyperboloid model [Nickel and Kiela, 2018], since it can avoid the numerical instabilities from the fraction in distance function, hence allows for more efficient computation on the manifold. We provide the empirical evaluation of the two hyperbolic models in Section 5.5.2.

## B. Proofs of Section 4.2 about Manifold Alignment

### B.1 The Loss Function of Manifold Alignment

As presented in Section 4.2, the loss function includes two parts, one for preserving the local manifold within each dataset, another for recovering the correspondence information.

We can rewrite $\mathcal{L}_M$ using the joint adjacency matrix:

$$\mathcal{L}_M = \sum_{i,j} \|\Phi_i - \Phi_j\|^2 J_{ij}, \tag{10}$$

where $\Phi$ is the unified representation of the taxonomy entities and the knowledge graph entities, $\Phi = \begin{bmatrix} \phi_{I_{\mathcal{T}}}(\mathbf{I}^{\mathcal{T}}) \\ \phi_{I_{\mathcal{G}}}(\mathbf{I}^{\mathcal{G}}) \end{bmatrix}$. $J$ is the joint adjacency matrix, $J = \begin{bmatrix} \mu S^{\mathcal{T}} & (1-\mu)W \\ (1-\mu)W^T & \mu S^{\mathcal{G}} \end{bmatrix}$.

Eq. (10) is the loss function for Laplacian eigenmaps [Belkin and Niyogi, 2003], which can be further derived into:

$$\mathcal{L}_M = \sum_{i,j} \sum_k (\Phi_{i,k} - \Phi_{j,k})^2 J_{ij} = \sum_k \sum_{i,j} (\Phi_{i,k} - \Phi_{j,k})^2 J_{ij} = \sum_k \text{tr}(\Phi_{\cdot,k}^T L \Phi_{\cdot,k}) = \text{tr}(\Phi^T L \Phi), \tag{11}$$

where the Laplacian $L = D - J$ and $D$ is a diagonal matrix with $D_{ii} = \sum_j J_{ji}$.

It is easily seen that mapping all entities to zero ($\Phi = \mathbf{0}$) minimizes the loss function $\mathcal{L}_M$, so an additional constraint $\Phi^T D \Phi = I$ needs to be added, where $I$ is the identity matrix.

**Theorem 1.** *Minimizing $\mathcal{L}_M$ with the constraint $\Phi^T D \Phi = I$ to get the d-dimensional aligned embeddings (a $(|I_{\mathcal{T}}| + |I_{\mathcal{G}}|) \times d$ matrix $\Phi$) is equivalent to solving a generalized eigenvalue problem of the joint graph Laplacian: $L\mathbf{v} = \lambda D \mathbf{v}$.*

*Proof.* When $d = 1$, $\Phi$ is a $(|I_{\mathcal{T}}| + |I_{\mathcal{G}}|)$-vector, denoted as $\mathbf{v}_1$, then we have

$$\arg \min_{\mathbf{v}_1 : \mathbf{v}_1^T D \mathbf{v}_1 = 1} \mathcal{L}_M = \arg \min_{\mathbf{v}_1, \lambda : \lambda > 0} \mathbf{v}_1^T L \mathbf{v}_1 + \lambda (1 - \mathbf{v}_1^T D \mathbf{v}_1). \tag{12}$$

Differentiating Eq. (12) with respect to $\mathbf{v}_1$ and $\lambda$, we have

$$L\mathbf{v}_1 = \lambda D \mathbf{v}_1, \tag{13}$$

$$\mathbf{v}_1^T D \mathbf{v}_1 = 1. \tag{14}$$

So the optimal $\mathbf{v}_1$ is a solution of the generalized eigenvalue problem: $L\mathbf{v}_1 = \lambda D \mathbf{v}_1$. Multiplying both sides of Eq. (13) by $\mathbf{v}_1^T$ and applying Eq. (14), we have $\mathbf{v}_1^T L \mathbf{v}_1 = \lambda$. To minimize $\mathcal{L}_M = \mathbf{v}_1^T L \mathbf{v}_1$, we need to get the smallest nonzero eigenvalue $\lambda_1$, then its corresponding eigenvector $\mathbf{v}_1$ is the optimal solution.

When $d > 1$, $\Phi$ can be written as $\Phi = [\mathbf{v}_1, \mathbf{v}_2, \ldots, \mathbf{v}_d]$. The optimization problem is

$$\arg \min_{\Phi : \Phi^T D \Phi = I} \mathcal{L}_M = \arg \min_{\mathbf{v}_1, \ldots, \mathbf{v}_d, \lambda_1, \ldots, \lambda_d : \lambda_i > 0} \sum_i \mathbf{v}_i^T L \mathbf{v}_i + \lambda_i (1 - \mathbf{v}_i^T D \mathbf{v}_i). \tag{15}$$

Applying the same technique above, we have the minimal $\mathcal{L}_M = \sum_i \lambda_i$, where $\lambda_i$, for $i = 1, \ldots, d$, are the $d$ smallest nonzero eigenvalues sorted in ascending order: $0 < \lambda_1 \leq \cdots \leq \lambda_d$, while their corresponding eigenvectors $\mathbf{v}_1, \ldots, \mathbf{v}_d$ are the optimal solutions. $\qquad\square$

### B.2 The Symmetry of $J$

**Theorem 2.** *The joint adjacency matrix* $J = \begin{bmatrix} \mu S^{\mathcal{T}} & (1 - \mu)W \\ (1 - \mu)W^T & \mu S^{\mathcal{G}} \end{bmatrix}$ *is symmetric.*

*Proof.* Recall the construction of the adjacency matrices $A^{\mathcal{T}}$ and $A^{\mathcal{G}}$. $A_{ij}^{\mathcal{T}} = 1$ if and only if entity $i$ is among the $k$ nearest neighbors of entity $j$ or $j$ is among the $k$ nearest neighbors of $i$ according to $\mathbf{I}^{\mathcal{T}}$, and the same with $A^{\mathcal{G}}$. Thus $A^{\mathcal{T}}$ and $A^{\mathcal{G}}$ are symmetric.

Then the similarity matrices are constructed through Eq. (4). We have $S_{ij}^{\mathcal{T}} = \exp(-\frac{d_h(\mathbf{I}_i^{\mathcal{T}}, \mathbf{I}_j^{\mathcal{T}})}{t}) \cdot A_{ij}^{\mathcal{T}} = \exp(-\frac{d_h(\mathbf{I}_i^{\mathcal{T}}, \mathbf{I}_j^{\mathcal{T}})}{t}) \cdot A_{ji}^{\mathcal{T}} = S_{ji}^{\mathcal{T}}$, similarly $S_{ij}^{\mathcal{G}} = S_{ji}^{\mathcal{G}}$, so $S^{\mathcal{T}}$ and $S^{\mathcal{G}}$ are symmetric. Then $J$ is also symmetric. $\qquad\square$

### B.3 The Symmetry and Positive Semi-Definiteness of $L$

**Theorem 3.** *The Laplacian* $L = D - J$ *is symmetric and positive semidefinite.*

*Proof.* First, $L$ is symmetric because $D$ and $J$ are both symmetric ($D$ is a diagonal matrix and $J$ is proved in Appendix B.2).

Next, from Eqs. (10) and (11), we have $\sum_{i,j} \|\Phi_i - \Phi_j\|^2 J_{i,j} = \mathrm{tr}(\Phi^T L \Phi)$, which shows that $L$ is positive semi-definite. $\qquad\square$

### B.4 The Solutions of Manifold Alignment

**Theorem 4.** *The generalized eigenvalue problem* $L\mathbf{v} = \lambda D \mathbf{v}$ *has a zero eigenvalue* $\lambda_0 = 0$.

*Proof.* Since $L = D - J$, $D_{ii} = \sum_j J_{ji} = \sum_j J_{ij}$,

then $(L \cdot [1, 1, \ldots, 1]^T)_i = \sum_j L_{ij} = D_{ii} - \sum_j (J_{ij}) = 0$,

that is, $L \cdot [1, 1, \ldots, 1]^T = \mathbf{0}$, so $\lambda_0 = 0$. $\qquad\square$

**Remark.** *Combining Appendix B.1 and the fact that $L$ is positive semi-definite (Appendix B.3), the solutions of manifold alignment using Laplacian eigenmaps are $\mathbf{v}_1, \ldots, \mathbf{v}_d$, which are the corresponding eigenvectors of the $d$ smallest nonzero eigenvalues: $0 = \lambda_0 < \lambda_1 \leq \ldots \lambda_d$.*
*Recall that $\Phi = \begin{bmatrix} \phi_{I_\mathcal{T}}(\mathbf{I}^\mathcal{T}) \\ \phi_{I_\mathcal{G}}(\mathbf{I}^\mathcal{G}) \end{bmatrix} = [\mathbf{v}_1, \mathbf{v}_2, \ldots, \mathbf{v}_d]$, so the aligned embedding vectors for $I_\mathcal{T}$ and $I_\mathcal{G}$ in the shared manifold are*

$$\phi_{I_\mathcal{T}}(\mathbf{I}_i^\mathcal{T}) = (\mathbf{v}_1(i), \ldots, \mathbf{v}_d(i)), \quad 1 \leq i \leq |I_\mathcal{T}|,$$
$$\phi_{I_\mathcal{G}}(\mathbf{I}_j^\mathcal{G}) = (\mathbf{v}_1(|I_\mathcal{T}| + j), \ldots, \mathbf{v}_d(|I_\mathcal{T}| + j)), \; 1 \leq j \leq |I_\mathcal{G}|.$$

## C. $\delta$-Hyperbolicity

The $\delta$-hyperbolicity of a graph $G$ [Gromov, 1987] is defined as follows.

**Definition 1.** *Let $a, b, c, d$ be vertices of the graph $G$. Let $S_1$, $S_2$ and $S_3$ be*

$$S_1 = dist(a, b) + dist(d, c),$$
$$S_2 = dist(a, c) + dist(b, d),$$
$$S_3 = dist(a, d) + dist(b, c).$$

*Suppose $M_1$ and $M_2$ are the two largest values among $S_1$, $S_2$, $S_3$ and $M_1 \geq M_2$. Define $hyp(a, b, c, d) = M_1 - M_2$. Then **the $\delta$-hyperbolicity** of $G$ is defined as*

$$\delta(G) = \frac{1}{2} \max_{a,b,c,d \in V(G)} hyp(a, b, c, d).$$

*That is, $\delta(G)$ is the maximum of hyp over all possible 4-tuples $(a, b, c, d)$ divided by 2.*

## D. More Experiments

### D.1 Hyperparameters

For pretraining of our approach, we learn the $n$-dimensional hyperbolic embeddings of the training taxonomy and the $n$-dimensional TransE embeddings of YAGOfacts, for $n = \{5, 10, 20, 50, 100\}$. In manifold alignment, we project the embeddings of the two entity sets into 10-dimensional Euclidean space. We use 5 nearest neighbors to construct $A^\mathcal{T}$ and $A^\mathcal{G}$. We assume the heat kernel parameter in Eq. (4) to be $+\infty$, i.e., $S^\mathcal{T} = A^\mathcal{T}, S^\mathcal{G} = A^\mathcal{G}$. The coefficient $\mu$ in $\mathcal{L}_M$ Eq. (5) is set as 0.25. The number of neighbors in $k$-nn when connecting the taxonomy entities $I_\mathcal{T}$ and knowledge graph entities $I_n$ is 5. The weights of edges in $E_n$ for training the new taxonomy are 0.05. For pretraining and retraining, the epochs of hyperbolic embeddings and TransE are 500.

The hyperparameters of baselines are also tuned on the knowledge base combining YAGOwordnet and YAGOfacts in 50-dimension by grid search, which are given in Table

| Model | Hyperparameters |
|---|---|
| TransE | sampling: *normal*; bern: *False*; regularize rate: 1; learning rate: 1; method: *sgd*; loss: *margin loss*; margin in loss: 5; # negative entity: 25; epochs: 500. |
| ComplEx | sampling: *normal*; bern: *True*; regularize rate: 1; learning rate: 0.5; method: *adagrad*; loss: *softplus loss*; # negative entity: 25; epochs: 1000. |
| RotatE | sampling: *cross*; bern: *False*; regularize rate: 0; learning rate: $2e^{-5}$; method: *adam*; loss: *sigmoid loss*; adv temp: 2; # negative entity: 25; margin: 6; epsilon: 2; epochs: 1000. |
| TransC | bern: *False*; learning rate: 0.01; margin: 1; instance margin: 0.4; concept margin: 0.3; epochs: 500. |
| HAKE | learning rate: 0.01; gamma: 24; alpha: 2; modulus weight: 3.5; phase weight: 1.0; #negative entity: 25; epochs: 180000. |
| JOIE | base model: *TransE*; bridge: *CMP-linear*; instance learning ratio: 2.5; fold: 3; concept learning ratio: 1.0; KG learning rate: 0.5; tax learning rate: 1.0; epochs: 500. |
| MurP | learning rate:50; #negative entity: 50; model: *poincare*; epochs: 500. |
| AttH | learning rate: 0.0005; regularizer: *N3*; regularization weight: 0; optimizer: *Adam*; # negative entity: 50; dropout: 0; gamma: 0; bias: *learn*; multi curvature: *True*; epochs: 500. |
| HyperKA | learning rate: 0.001; instance layer num: 3; ontology layer num: 3; negtive typing margin: 0.1; epsilon: 1.0; negtive triple margin: 0.2; # negative entity: 20; mapping: *True*; combine: *True*; epochs: 500. |

Table 8: Hyperparameters of the baselines. Tuned on the knowledge base combining YAGOwordnet and YAGOfacts in 50-dimension.

8. We use the OpenKE repository[2] to train TransE, ComplEx, and RotatE. For TransC,[3] HAKE,[4] JOIE,[5] MurP[6], AttH[7] and HyperKA[8], we use their public codes.

## D.2 Evaluation

Our evaluation closely follows the setting of [Nickel and Kiela, 2017, 2018], which infers the hierarchies from distances in the embedding space. Specifically, for each test entity $u$ and the ground truth edge $(u, v)$, we compute the distance between the embeddings $d_h(\mathbf{u}, \mathbf{v})$ and rank it among the distances of all unobserved edges for $u$: $\{d_h(\mathbf{u}, \mathbf{v}') : (u, v') \notin \text{Training}\}$. We then report the following evaluation metrics of the rankings. Denote $I_{test}$ as the test entity set. Let $NE_u = \{v_1, v_2, \ldots, v_{|NE_u|}\}$ be the set of the ground truth types of entity $u$.

---

2. https://github.com/thunlp/OpenKE
3. https://github.com/davidlvxin/TransC
4. https://github.com/MIRALab-USTC/KGE-HAKE
5. https://github.com/JunhengH/joie-kdd19
6. https://github.com/ibalazevic/multirelational-poincare
7. https://github.com/HazyResearch/KGEmb
8. https://github.com/nju-websoft/HyperKA

|  | Test entity | Neighbors | Prediction |
|---|---|---|---|
| TransE | Eric Clapton | Brian Eno (artist)
Smokey Robinson (artist)
Chet Atkins (artist) | **organism**
**causal agent**
**living thing** |
| GeoAlign | Eric Clapton | James Brown (artist)
Chet Atkins (artist)
Willie Nelson (artist) | **artist***
**creator**
**person** |
| TransE | Neil Young | Tony Banks (artist)
Robbie Williams (artist)
Glen Campbell (artist) | **organism**
**causal agent**
property |
| GeoAlign | Neil Young | Moby (artist)
Elton John (artist)
Björn Ulvaeus (artist) | **creator**
**artist***
**person** |
| TransE | Neil Gaiman | Stephen King (21st-century American novelists)
Gene Wolfe (fantasy writers)
Jonathan Lethem (American male essayists) | American male writers
American novelists
20th-century American novelists |
| GeoAlign | Neil Gaiman | Nikolai Gogoln (mythopoetic writers)
Drago Jančar (Slovenian novelists)
Charles Stross (Scottish science fiction writers) | **North American writers***
**non-fiction writers***
**English screenwriters*** |
| TransE | Hannah Arendt | Judith Butler (20th-century women writers)
Walter Benjamin (Franz Kafka scholars)
Karl Jaspers (Philosophers of technology) | Translators
American novelists
American fiction writers |
| GeoAlign | Hannah Arendt | Milan Kundera (21st-century French novelists)
Horace (Golden Age Latin writers)
Imre Lakatos (20th-century Hungarian writers) | **People in literature**
**Writers**
**European writers*** |

Table 9: Examples of nearest neighbors and top 3 predictions for the test entity. The labels of the neighbors are provided in parentheses. The correct predictions are bold-faced. The fine-grained correct predictions are marked with *.

**Mean average precision (MAP).** The average precision (AP) is a way to summarize the precision-recall curve into a single value representing the average of all precisions and the MAP score is calculated by taking the mean AP over all classes. For an entity $u$, from the learned embeddings, we can obtain the types closest to its embedding $\mathbf{u}$. Let $R_{u,v_i}$ be the smallest set of such types that contains $v_i$ (the $i$-th ground truth type of $u$). Then the MAP is defined as:

$$\text{MAP} = \frac{1}{|I_{test}|} \sum_{u \in I_{test}} \frac{1}{|NE_u|} \sum_{v_i \in NE_u} Precision(R_{u,v_i}).$$

**Mean reciprocal rank (MRR).** The MRR is a statistic measure for evaluating a list of possible responses to a sample of queries, ordered by the probability of correctness. For an entity $u$, from the learned embeddings, we can rank its distances with the types from the smallest to the largest. Let $rank_{v_i}$ be the rank of $v_i$ (the $i$-th ground truth type of $u$). Then the MRR is defined as:

$$\text{MRR} = \frac{1}{|I_{test}|} \sum_{u \in I_{test}} \frac{1}{|NE_u|} \sum_{v_i \in NE_u} \frac{1}{rank_{v_i}}.$$

**The proportion of correct types that rank no larger than $N$ (Hits@$N$).** Hits@$N$ measures whether the top $N$ predictions contain the ground truth labels. For an entity $u$, from the learned embeddings, we can obtain the set of $N$ types closest to its embedding $\mathbf{u}$, denoted as $R_u^N$. Then the Hits@$N$ is defined as:

$$\text{Hits@}N = \frac{1}{|I_{test}|} \sum_{u \in I_{test}} \mathbb{I}(|R_u^N \cap NE_u| \geq 1),$$

where $\mathbb{I}(|R_u^N \cap NE_u| \geq 1)$ is the indicator function.

## D.3 Case Study

Our manifold alignment is on top of the pretrained TransE model and the hyperboloid embeddings. To have a more intuitive sense about the manifold alignment, in Table 9, we give some examples of the nearest neighbors of the test entity in the aligned manifold as well as the top 3 types predicted by GeoAlign. We also present the corresponding items of TransE as a comparison. For TransE, the nearest neighbors are from its embedding space. The examples *Eric Clapton* and *Neil Young* are from YAGOwordnet while *Neil Gaiman* and *Hannah Arendt* are from wikiObjects. As we expect, TransE and manifold alignment which leverages TransE both find the reasonable nearest neighbors. When looking into the top 3 predictions of *Eric Clapton* and *Neil Young* , we see that although TransE predicts correctly, it cannot get the fine-grained types. The most confident predictions of TransE are usually the general and rough types such as *organism, living thing*, while GeoAlign successfully predict them as *artist*. For the test entities *Neil Gaiman* and *Hannah Arendt* , which come from the more fine-grained and massive taxonomy wikiObjects, TransE cannot predict correctly at the top 3 predictions, but GeoAlign successfully obtains the correct fine-grained types. The results show that GeoAlign not only outputs accurate predictions but also captures the hierarchy structure to some extent.

