# OpenReview forum: "Manifold Alignment across Geometric Spaces for Knowledge Base Representation Learning"
_AKBC.ws/2021/Conference — AKBC 2021_

### Official Review · Reviewer_ggEQ · 2021-07-21
**Interesting idea, solid results, good analysis**

**Rating:** 7
**Confidence:** 4

**Review:**

The authors propose a knowledge base embedding method which learns representations through manifold alignment across geometric spaces rather than relying solely on the Euclidean or the hyperbolic space. As a result, the representations can capture multiple relation properties found in knowledge bases, thus overcoming the limitations of existing approaches. Empirical evaluation on an out-of-taxonomy entity typing task demonstrates that the method is effective and beats a plethora of well-known competitors.

### Strengths:

* Overall, the paper is well written, easy to follow, and looks technically correct. The motivation and goals are quite clear. The claims are well supported by theories and their derivations, and it includes an adequate amount of citations.

* The main contribution of the paper is a knowledge base embedding method that can support relation properties such as symmetry, inversion, composition, and transitivity, whereas previous methods were only able to support a subset. The method is sound (with little added complexity in comparison to other methods) and clearly presented. It achieves robust results in comparison to other methods in the literature.

### Weaknesses and Questions:

1. My biggest concern is that the empirical analysis is lacking due to only using two datasets. Given that the performance varies across the two, it would seem natural to add at least one more dataset to further confirm the observed performance trends.

2. In subsection 5.2 you say “... do various knowledge base inferences instead of restricted in the hyperbolic embeddings”. This needs more justification and discussion.

3. It would have been nice to see a thorough discussion about when this method can be used in practise and when not, and also present some experimental results of the limitations (e.g. error analysis).

4. The analysis and the contents of Table 9 seem anecdotal. If the authors want to conduct a case study, presenting only 2 examples is not enough.

5. How does this method compare to other competitors (e.g., MurP, AttH, etc.) in terms of runtime?

6. In subsection 5.2 you say “	We think the neural models may not fit this task and our datasets”. Have you considered whether the out-of-taxonomy entity typing task is particularly advantageous to your method, while disadvantageous to the baselines? Comparing the state-of-the-art baselines with tasks they used (e.g. reconstruction) will significantly strengthen the evaluation.

7. **Minor Comment** Even though the paper is well written, my overall feeling is it tries to compress too much information into 10 pages. I believe the text spends too much time on the background and too little on the experiments.

---

> ### Author Response · Authors · 2021-07-30
> **Response to Reviewer ggEQ**
>
> We would like to thank the reviewer for the efforts and the constructive comments. In the following, we address the weaknesses and questions provided by the reviewer.
>
> 1. Thanks for your suggestion. We will add more datasets for empirical analysis in our final version.
> 2. In Section 5.2, we mentioned that ''The flexibility enables our approach to do various knowledge base inferences instead of restricted in the hyperbolic embeddings.'' because GeoAlign can take any base embedding models rather than TransE, which enables our approach to take advantage of the well-developed Euclidean embedding models as our pretraining model for the knowledge graph, while the hyperbolic emebdding models (MurP and AttH) can only do the inference for all relation properties in the hyperbolic space, which is not suitable for the non-transitive relation properties. Thanks for for pointing out the unclarity. We added the corresponding discussion in our revision.
> 3. The good performance on the out-of-taxonomy entity typing task demonstrates that our approach is very useful to predict the types of the new entities for taxonomies in practice, even when the taxonomy is small. However, as we presented in Section 5, when the training rate or the dimension is sufficiently large, our improvements are limited compared with MurP.
> 4. Thanks for the suggestion, we added two more examples in Table 9 in our revision. We will work on more examples for the case study in the final version.
> 5. In terms of runtime, the pretraining and retraining which use TransE and the hyperboloid embeddings are very efficient, having few difference with other baselines. However, since we need to run the hyperboloid embedding model twice and we also need to run the manifold alignment algorithm (which takes 5min$\sim$10min, depending on the size of the dataset), our approach costs more runtime.
> 6. The neural model HyperKA deals with two tasks: entity alignment and type inference, where the type inference is in the similar setting with our experiments. As we specified in Section 2, we focus on a different topic with the entity alignment. As for the reconstruction task, the hyperbolic embedding models including the Poincare ball model and the hyperboloid model achieve the advanced results. By comparing with them, we showed that our approach has acceptable compromise in the reconstruction task in the ablation study.
> 7. Thanks for the suggestion. Since the revision allows for one extra page, we put the experiments of the ablation study in the main body of the paper and shortened the background on the related work. We hope this version will look better-organized and more balanced between the background and the experiments.

---

### Official Review · Reviewer_qKib · 2021-07-22
**Well-motivated models received good results, but needs more verification.**

**Rating:** 7
**Confidence:** 4

**Review:**

**Paper Summary**:

This paper proposed a framework to align Euclidean space knowledge base completion models such as TransE, and hyperbolic space model that is specifically great at modeling hierarchical relations to a common space. In the knowledge base completion task, there are both non-hierarchy relations as well as hierarchical relations, and they can be modeled better in different geometric spaces. The proposed model tried to align different spaces that are good at different type relations into a common space to have the benefits of both. The authors performed experiments on two different datasets and showed improved performance compared to multiple strong baselines.

**Strength**:
- The proposed model is intuitive and well-motivated. The mathematical motivation is strong, and the authors explain the models well. The paper is also well-written.
- The proposed model achieved better results compared to multiple strong baselines in the proposed missing entity type prediction task.
- The proposed model is flexible, and the KB component inside the framework can be replaced with other KB models other than TransE, which is used in the paper.

**Weakness / Questions / Suggestions**:
- The baseline and the proposed model parameter are not well-specified. There should be more details of how the parameters are choosing, and how do the models get tuned for each model to make sure it is a fair comparison between models.
- For the smaller learning rate analysis in section 5.4, why should we pay attention to smaller learning rate results? What does it mean practically? On the other hand, the performance using different learning rates is tied with the batch size. What is the batch size used in these models? Are they all the same for all the models?
- At the end of the abstract, the authors claimed that the proposed approach achieved significantly good performance compared to other models, but the performance of GeoAlign is very close to MurP in table 3. So it would be more convincing if you can do the significance testing of the claim you made in the abstract.
- The experiment in this paper is the missing entity type prediction, where the proposed model showed improved results. But it remains unclear for the main task performance, i.e., the knowledge base completion task (after retraining with the new entities in section 4.3). This is especially important when the paper title is ".... for knowledge base representation learning," in which case "knowledge base representation" includes more aspects than missing entity type predictions.

---

> ### Author Response · Authors · 2021-07-30
> **Response to Reviewer qKib**
>
> We would like to thank the reviewer for the efforts and the valuable comments. In the following, we address the questions of the reviewer.
>
> * __Parameter tuning.__ As specified in Section 5.1.3, for all methods, we tune the hyperparameters on the knowledge base combining YAGOwordnet and YAGOfacts by grid search according to the MAP score on the out-of-taxonomy entity typing task, i.e., the knowledge base, the test set, and the evaluation metric used for hyperparameter tuning are the same for all methods. The hyperparameters are given in Appendix D.1.
> * __Small training rate analysis.__ The real-world taxonomies may contain limited entities, which corresponds to the small learning rate setting. Our manifold alignment framework can fulfill the taxonomy completion task by linking the new entities from the massive knowledge graph to the original taxonomy. When the real-world taxonomy contains very few entities, our framework can be combined with the semi-supervised learning methods or the bootstrap methods to deal with the few-shot learning. That would be an interesting and potential future work.
>
>     The hyperparameters including the learning rate and batch size remain the same on small training rates. And the hyperparameter tuning is addressed in the previous point.
> *__Significance testing.__ Thanks for the suggestion. In our revision, we added the significance testing in our experiments and reported the statistically significance metrics in Table 2, 3, and 4 regarding the concern about close performance with MurP.
> * __Knowledge base completion task.__ Our manifold alignment framework mainly focuses on the out-of-taxonomy entity typing task. As for the inference for other non-transitive relation properties, the sota Euclidean models already achieved great results. Our motivation is to use different geometric spaces to best characterize various relation properties while aligning the underlying local structures of the shared entities.

---

### Official Review · Reviewer_rooA · 2021-07-22
**Interesting work, with solid experimentation and analyses**

**Rating:** 8
**Confidence:** 3

**Review:**

The paper presents a method to learn knowledge base (KB) embeddings in different geometric spaces (Euclidean, hyperbolic), which are good at representing certain relations (e.g., Euclidean: symmetry, inversion, composition) and weak for others (e.g., Euclidean: transitivity), then apply manifold alignment to align the shared entities, thus leveraging the benefits of both spaces. The work is motivated by limitations of prior work as well as the goal of building a representation framework for all relation properties. Experimental results on an out-of-taxonomy entity typing task (2 datasets) demonstrate the goodness of the approach, and its superiority compared to previous models (KB embedding models, hierarchy-aware Euclidean methods, multi-relational hyperbolic models).


### Strengths

- The paper is well-written, accompanied by a clear motivation and detailed background materials, derivations, and insights in the Appendix.
- The contribution is solid and the method is both conceptually simple and effective. The method is also clearly explained, with sufficient details and motivation.
- The empirical results are convincing, and the method achieves superior performance compared to previous approaches.


### Weaknesses and comments

- Experiments. Each running is executed 5 times and the mean values of results are reported. It would be great to have (in the Appendix) the individual results that are then averaged, and possibly the standard deviation in addition to mean values. This would shed light on the stability of the performance.
- There are interesting experiments on Appendix D that would fit better in the main body for the reader. My suggestion is to reserve space for (at least part of) this information in the main body (e.g., ablation study on Appendix D.3), putting part of the background Section 1-2-3 on the Appendix instead.
- Typo. Pag. 7: "YAGOfacts are used for preraining" -> "YAGOfacts are used for pretraining".

---

> ### Author Response · Authors · 2021-07-30
> **Response to Reviewer rooA**
>
> We would like thank the reviewer for the efforts and helpful comments. We made the following revisions according to the comments.
>
> * __Experiments.__ Instead of reporting the standard deviation, we added the significance testing in our experiments and reported the statistically significance metrics computed from the mean value and std.
> * __Main body and Appendix.__ Thanks for the suggestion. Since the revision allows for one extra page, we put the experiments of the ablation study in the main body of the paper and shortened the background on the related work. We hope this version will look better-organized and more balanced between the background and the experiments.
> * __Typo.__ Thanks for your carefulness. We fixed the typo in our revision.

---

### Author Response · Authors · 2021-07-30
**Response to all reviewers and Summary of the revisions**

We sincerely appreciate the comments and feedbacks given by the reviewers. We also appreciate the reviewers' interests in our work very much. Here we summarize the revisions in our updated version.

1. we put the experiments of the ablation study in the main body of the paper and shortened the background on the related work.
2. we added the significance testing in our experiments and reported the statistically significance metrics in Table 2, 3, and 4.
3. We added more explanations in Section 5.2 to avoid confusion.
4. We added two more examples in the case study in the Appendix.
5. We fixed some typos.

Thanks again for the suggestions. The suggestions and comments help us to improve our paper.

---

### Decision · Program_Chairs · 2021-08-17

**Decision:**

Accept

**Comment:**

This paper proposes to learn knowledge graph embeddings in different geometric spaces (Euclidean and hyperbolic space) to handle different types of relations, and the shared entities in different spaces are further aligned through manifold alignment. All the reviewers agree the paper is well written, the proposed method is intuitive and well explained, and the experiments are convincing.